Mitochondrial complex activity in permeabilised cells of chronic fatigue syndrome patients using two cell types

Tomas Cara 1
Brown Audrey E. 1
Newton Julia L. 1 2
Elson Joanna L. j.l.elson@ncl.ac.uk 3 4
1 Institute of Cellular Medicine, Newcastle University , Newcastle upon Tyne , United Kingdom
2 NHS Foundation Trust, Newcastle upon Tyne Hospitals , Newcastle upon Tyne , United Kingdom
3 Institute of Genetic Medicine, Newcastle University , Newcastle upon Tyne , United Kingdom
4 Centre for Human Metabonomics, North-West University , Potchefstroom , South Africa
Gillespie Joseph
Electronic publication date: 2019 Mar 1
Publication date: 2019
Volume: 7
Electronic Location ID: e6500
Received 2018 Oct 18; Accepted 2019 Jan 22
Copyright: ©2019 Tomas et al.
Copyright year: 2019
Copyright holder: Tomas et al.
License: This is an open access article distributed under the terms of the Creative Commons Attribution License, which permits unrestricted use, distribution, reproduction and adaptation in any medium and for any purpose provided that it is properly attributed. For attribution, the original author(s), title, publication source (PeerJ) and either DOI or URL of the article must be cited.
License URL: https://creativecommons.org/licenses/by/4.0/

Keywords: Myalgic encephalomyelitis, Mitochondrial, Peripheral blood mononuclear cells (PBMCs), Skeletal muscle (myotubes), OXPHOS

Funding: The Medical Research Council, Action for ME, ME research UK, and the ME Association The following funding bodies provided funding for this study: The Medical Research Council, Action for ME, ME research UK, and the ME Association. The funders had no role in study design, data collection and analysis, decision to publish, or preparation of the manuscript.

==============================
Abnormalities in mitochondrial function have previously been shown in chronic fatigue syndrome (CFS) patients, implying that mitochondrial dysfunction may contribute to the pathogenesis of disease. This study builds on previous work showing that mitochondrial respiratory parameters are impaired in whole cells from CFS patients by investigating the activity of individual mitochondrial respiratory chain complexes. Two different cell types were used in these studies in order to assess individual complex activity locally in the skeletal muscle (myotubes) (n = 6) and systemically (peripheral blood mononuclear cells (PBMCs)) (control n = 6; CFS n = 13). Complex I, II and IV activity and respiratory activitysupported by fatty acid oxidation and glutaminolysis were measured usingextracellular flux analysis. Cells were permeabilised and combinations of substrates and inhibitors were added throughout the assays to allow states of mitochondrial respiration to be calculated and the activity of specific aspects of respiratory activity to be measured. Results showed there to be no significant differences in individual mitochondrial complex activity or respiratory activity supported by fatty acid oxidation or glutaminolysis between healthy control and CFS cohorts in either skeletal muscle (p ≥ 0.190) or PBMCs (p ≥ 0.065). This is the first study to use extracellular flux analysisto investigate individual mitochondrial complex activity in permeabilised cells in the context of CFS. The lack of difference in complex activity in CFS PBMCs suggests that the previously observed mitochondrial dysfunction in whole PBMCs is due to causes upstream of the mitochondrial respiratory chain.

Introduction

Chronic fatigue syndrome (CFS), also known as Myalgic Encephalomyelitis (ME), is a debilitating disease affecting 0.2–0.4% of the population in the UK (NICE, 2007). CFS has a significant impact on the quality of life of patients with key symptoms including severe fatigue and post-exertional malaise (Hvidberg et al., 2015; Winger et al., 2015). The mechanisms behind the aetiopathogenesis of CFS are yet to be elucidated. The lack of knowledge of the mechanisms behind the disease contribute to difficulty in obtaining a consensus on diagnostic criteria and the development of widely effective treatments.

Various aspects of mitochondrial dysfunction have previously been postulated as contributing to CFS (Tomas et al., 2017; Booth, Myhill & McLaren-Howard, 2012; Myhill, Booth & McLaren-Howard, 2009; Myhill, Booth & McLaren-Howard, 2013; Lawson et al., 2016; Behan, More & Behan, 1991; Morris & Maes, 2014; Filler et al., 2014). Fatigue has been shown to be common in patients with primary mitochondrial disease (Gorman et al., 2015); however, it should be noted that patients with CFS are not seen to harbour primary mitochondrial mutations (Schoeman et al., 2017). Billing-Ross et al. (2016) showed that certain changes in the mitochondrial genome increases the likelihood of specific symptoms in CFS patients such as gastrointestinal, neurological, and inflammatory symptoms, however, these genomic changes do not make patients more susceptible to developing the disease. Previous studies have shown the energy production, including mitochondrial activity, of whole PBMCs from CFS patients to be significantly impaired compared to a healthy control cohort (Tomas et al., 2017; Fluge et al., 2016). One study, using extracellular flux analysis in whole cells, showed CFS PBMCs to have significantly impaired mitochondrial functioning both under basal conditions and when maximally stimulated to respire, under a number of experimental conditions (Tomas et al., 2017). This suggested that CFS patients were unable to utilise mitochondrial energy production to the same extent as healthy controls and implied that mitochondrial dysfunction may contribute to the pathogenesis of the disease. The study presented here was conducted in order to further investigate if the mitochondrial dysfunction seen in CFS PBMCs was due to atypical activity of individual mitochondrial complexes using the same technique as the previous study (extracellular flux analysis) but in permeabilsied cells rather than whole cells. The use of permeabilised cells allows mitochondria to be directly accessed by the substrates with no cellular interference in terms of substrate transport or intracellular interactions. The permeabilisation of cells also allows the enzymatic activity of individual respiratory chain complexes to be to be measured which cannot be done easily in whole cells. This study investigates the activity of individual complexes and components of the mitochondrial respiratory chain in myotubes and PBMCs—myotubes were used to investigate mitochondrial activity locally in the skeletal muscle, while PBMCs were used to investigate mitochondrial complex activity systemically. This was achieved by permeabilising cells ensuring that the mitochondria remained intact and using extracellular flux analysis to record oxygen consumption rate of cells following the serial injection of mitochondrial activity altering compounds. Respiratory parameters were calculated and compared between control and CFS derived muscle cells and PBMCs.

For this study, saponin was used to permeabilise the cell membrane. Saponin is a cell permeabilser which acts by forming complexes with cholesterol leading to a reduction in cell membrane integrity, while keeping mitochondrial membranes intact (Jamur & Oliver, 2010). When mitochondria are isolated from cells, the architecture and morphology of the mitochondria is altered (Bach et al., 2003; Mitra et al., 2009; Sarin et al., 2013; Hagenbuchner et al., 2013), but permeabilisation of the cell membrane allows the architecture and morphology of mitochondria to remain normal, an advantage over the use of isolated mitochondria as mitochondrial function has previously been shown to have a strong relationship with structure (Saks et al., 1998; Picard et al., 2011). Permeabilisation of the cell membrane allows the effect of substrates on mitochondrial activity to be comprehensively assessed by allowing endogenous substrates to be delivered to mitochondria (Clerc & Polster, 2012). The addition of different substrates and inhibitors alters mitochondrial respiration and allows the activity of individual components of mitochondrial respiration to be measured  (Salabei, Gibb & Hill, 2014). Originally described by Chance and Williams in 1955, mitochondrial respiratory activity can be measured in terms of several respiratory ‘states’ (Chance & Williams, 1955). State 3 respiration, state 4 respiration and respiratory control ratio (RCR) are often used as markers for mitochondrial respiratory activity. State 3 respiration is when mitochondria have a high concentration of ADP externally, and a high oxygen consumption rate and ATP synthesis thus producing a state whereby ADP stimulated respiration can be measured (Chance & Williams, 1955; Chance & Williams, 1956). State 4 respiration, on the other hand, is when mitochondria have a very low external ADP concentration, and little or no ATP synthesis due to the complete phosphorylation of ADP to ATP. The respiratory control ratio (RCR) is a measure of the coupling of ATP synthesis and electron flux and shows the capacity of mitochondria to synthesise ATP via the oxidation of respiratory substrates (Hill et al., 2012).

This work aimed to determine if the enzymatic activity of different complexes of the mitochondrial respiratory chain differed between CFS patients and healthy controls either locally, in the skeletal muscle, or systemically in PBMCs.

Materials and Methods

Study participants

CFS and control derived primary myoblasts were obtained from muscle biopsies of the vastus lateralis of CFS patients and healthy controls and processed and gifted by Dr Audrey Brown, Newcastle University. All CFS patients fulfilled the Fukuda diagnostic criteria and were recruited via the Newcastle NHS CFS Clinical Service at the Newcastle Hospitals NHS Foundation Trust (Fukuda et al., 1994).

Blood samples were obtained from patients fulfilling the Fukuda Diagnostic criteria for CFS after obtaining ethical approval from the National Research Ethics Committee North East—Newcastle & North Tyneside 2 (Fukuda et al., 1994). Samples from healthy controls were collected through the Institute of Cellular Medicine (Newcastle University) blood study after obtaining ethical approval from the National Research Ethics Committee North East—County Durham & Tees Valley. Samples were gathered after informed written consent was obtained.

Reagents

All reagents were obtained from Sigma Aldrich, UK unless otherwise stated.

Cell culture and preparation

Myotubes

Myoblasts were grown to passage 7 in Ham’s F10 medium (Scientific Laboratory Supplies, Nottingham, UK) (supplemented with 20% fetal bovine serum (FBS) (Life Technologies, UK), 2% chick embryo extract (Sera Labs International, Haywards Heath, UK), 2% penicillin-streptomycin, 1% amphotericin B). Cells were then seeded at a density of 3 × 103 per well into a 96-well seahorse plate (Agilent Technologies, Wokingham, UK) in quadruplicate, and differentiated into myotubes in differentiation medium (minimal essential media supplemented with 2% FBS, 1% penicillin-streptomycin and 1% amphotericin B). Experiments were performed after 7 days of differentiation. Differentiation was confirmed by observing the formation of long, multinucleated myotubes in alignment under the microscope.

PBMCs

PBMCs were separated using Histopaque® as described by Tomas et al. (2017). The PBMCs used in these experiments were frozen at −80 °C in freezing medium (50% RPMI-1640, 40% FBS, 10% DMSO) and revived and plated the day before experiments. Wells of a 96-well seahorse plate were coated with poly-D-lysine, to aid in the attachment of cells, and left to air-dry for 2 hours prior to the plating of cells. Following revival of cells, PBMCs were seeded at a density of 5  × 105 cells per well in quadruplicate in the poly-D-lysine coated 96-well seahorse plate and incubated overnight in RPMI-1640 (supplemented with 10% FBS and 1% penicillin-streptomycin) at 37 °C and 5% CO2.

Extracellular flux analysis

The XFe96 analyser (Agilent Technologies) was used to investigate the activity of individual mitochondrial respiratory chain complexes using specific substrates. The protocol used in this study is described by Salabei, Gibb & Hill (2014) and the mix, wait and measure times provided by Agilent Technologies (0.5 min/0.5 min/2 min) (Agilent Technologies, 2016). Seeding densities for PBMCs and myotubes were used as described previously  (Tomas et al., 2017; Rutherford, 2016). Myotubes were seeded at a density of 3  × 103 cells per well while PBMCs were seeded at 5  × 105 per well. On the day of experiments, experimental medium was prepared by supplementing DMEM with 1mM pyruvate, 2 mM L-glutamine and 1 mM glucose. The pH of the media was adjusted to 7.4 with 0.1M NaOH and warmed to 37 °C. One hour before running the experiment, media was removed from each well of the XFe96 and replaced with 180 µl of prepared medium and incubated for one hour at 37 °C with no CO2. Mannitol and Sucrose (MAS) buffer (70 mM sucrose, 220 mM mannitol, 10 mM potassium phosphate monobasic, 5 mM magnesium chloride, 2 mM HEPES, 1 mM EGTA) was prepared. A 4 mg/ml fatty acid free bovine serum albumin (BSA) solution was created by adding BSA to MAS to create MAS-BSA buffer. The medium on the plate was replaced with 180 µl of MAS-BSA 10 min prior to the plate being loaded into the machine. Oxygen consumption rate (OCR) of cells was measured at 12 points throughout the assay. Three basal readings were made before the first injection containing a mix of the substrate(s) of interest, ADP, FCCP and saponin. Three subsequent readings were made and then the second injection, containing oligomycin, was added to the cells. Another three readings of OCR were made and the final injection of either rotenone or potassium azide was added to the cells, and a final three OCR readings recorded. Saponin concentration was optimised independently for myotubes and PBMCs and the damage to mitochondria caused by saponin was also assessed using cytochrome C (Data S1). The optimal concentration of saponin for myotubes was determined to be 25 µg/ml, while the optimum saponin concentration for PBMCs was 2.5 µg/ml. Data were normalised for protein concentration following a bicinchoninic acid (BCA) assay (Fisher Scientific, Loughborough, UK) conducted according to manufacturer’s instructions.

Compound preparation

Compounds and inhibitors used to investigate the activity of different complexes in the mitochondrial respiratory chain are shown in Table 1.

Table 1 Table showing the compounds used to investigate mitochondrial complex activity.

Compound(s)	10× port solution concentration	Final in well concentration	Complex investigated	
Pyruvate/malate	50 mM/25 mM	5 mM/2.5 mM	Complex I mediated respiration	
Succinate	100 mM	10 mM	Complex II mediated respiration	
Tetamethylphenylendiamine (TMPD)/Ascorbate	5 mM/20 mM	0.5 mM/2 mM	Complex IV activity	
Palmitoyl-l-carnitine	0.5 mM	50 µM	Respiratory activity supported by fatty acid oxidation	
Glutamine/malate	40 mM/5 mM	4 mM/0.5 mM	Respiratory activity supported by glutaminolysis	
ADP	10 mM	1 mM	State 3 respiratory activity	
Saponin	250 µg/ml	25 µg/ml	Cell permeabilisation	
FCCP	10 µM	1 µM	Maximal respiratory activity	
Oligomycin	10 µM	1 µM	State 4 respiratory activity	
Rotenone	10 µM	1 µM	Metabolic inhibitor	
Potassium Azide	200 mM	20 mM	Metabolic inhibitor	

Parameter calculations

For respiratory chain complex activity, state 3 respiration, state 4 respiration, respiratory control ratio (RCR), basal respiration and maximal respiration were calculated as shown below using the measurement numbers shown in Fig. 1. State 3 respiration=average of measurements 4–6−average of measurements 10–12

State 4 respiration=average of measurements 7–9−average of measurements 10–12

RCR=State 3 respirationstate 4 respiration.

Figure 1 Profile of the parameters of mitochondrial respiration measured in isolated mitochondria.

Data analysis

Groups were compared using student’s t-tests after confirming equal variances using Levene’s test for equality of variances. All statistical tests were carried out using IBM SPSS Statistics 22. Graphs were created using Graphpad Prism 7.

Results

Myotube respiratory chain complex activity

Myotube respiratory chain activity was analysed for state 3 respiration, state 4 respiration, and RCR with the addition of various combinations of substrates and inhibitors. The results for myotubes for complex II respiration (succinate), and complex IV activity (TMPD & ascorbate) are shown in Fig. 2. There were no significant differences between control (n = 6) and CFS (n = 6) cohorts for state 3 respiration, state 4 respiration, or RCR when cohorts were compared with student’s t-tests (p ≥ 0.190).

Figure 2 State 3 respiration, state 4 respiration, and RCR in control and CFS permeabilised myotubes.

Succinate was used to analyse complex II mediated respiration and (A) state 3 respiration, (B) state 4 respiration, and (C) RCR were measured. TMPD & ascorbate were added to investigate complex IV activity and (D) state 3 respiration, (E) state 4 respiration, and (F). RCR were measured. Groups were compared using student’s t-tests. Control n = 6; CFS n = 6.

Attempts were made to measure complex I mediated respiration, using pyruvate and malate; respiratory activity supported by fatty acid oxidation, with the addition of palmitoyl-l-carnitine; and respiratory activity supported by glutaminolysis, with the addition of glutamine and malate. Results for all three experiments consistently produced negative values for OCR in both CFS (n = 6) and control (n = 6) myotubes therefore we were unable to calculate state 3, state 4, and RCR for these experiments (Fig. 3).

Figure 3 Permeabilised myotube mitochondrial stress test traces.

Mitochondrial stress test traces in myotubes permeabilised with the addition of different substrates in the first injection alongside saponin, ADP and FCCP. (A) Glutamine and malate. (B) Palmitoyl-l-carnitine. (C) Pyruvate and malate. Control n = 6; CFS n = 6.

PBMC respiratory chain complex activity

PBMCs from healthy controls and CFS patients were used to investigate different aspects of mitochondrial respiratory chain activity outlined in Table 1. Results showed there to be no difference between state 3 respiration, state 4 respiration or RCR of PBMCs from CFS (n = 13) patients and healthy controls (n = 6; succinate controls n = 4) for any of the substrate/inhibitor combinations (p ≥ 0.065) (Fig. 4).

Figure 4 State 3 respiration, state 4 respiration, and RCR in control and CFS permeabilised PBMCs.

Succinate was used to analyse complex II mediated respiration and (A) state 3 respiration, (B) state 4 respiration, and (C). RCR were measured. TMPD & ascorbate were added to investigate complex IV activity and (D) state 3 respiration, (E) state 4 respiration, and (F) RCR were measured. Glutamine & malate allowed respiratory activity supported by glutaminolysis to be measured and (G) state 3 respiration, (H) state 4 respiration, and (I) RCR were recorded. Pyruvate & malate were used to investigate complex I mediated respiration and (J) state 3 respiration, (K) state 4 respiration, and (L) RCR were measured. Palmitoyl-l-carnitine was added to assess respiratory activity supported by fatty acid oxidation and (M) state 3 respiration, (N) state 4 respiration, and (O) RCR were recorded. Groups were compared using student’s t-tests. Control n = 6; CFS n = 13.

Discussion

The activity of different aspects of mitochondrial respiratory chain function were investigated by adding various combination of substrates and inhibitors. The effect of the different substrates on state 3 respiration, state 4 respiration and respiratory control ratio (RCR) was measured.

Five different combinations of substrates were investigated for their effects on myotubes and PBMCs—glutamine and malate; palmitoyl-l-carnitine; pyruvate and malate; succinate; TMPD and ascorbate. These were added to investigate respiration supported by glutaminolysis; respiration supported by fatty acid oxidation; complex I activity; complex II activity; and complex IV activity, respectively. Despite there being a lack of difference in OXPHOS between control and CFS cohorts shown in whole myotubes (G Rutherford, pers. obs., 2016)  (Rutherford, 2016), these experiments aimed to investigate if more subtle and specific differences occurred in individual complexes. PBMCs were used to see if the specific location of abnormalities identified in whole cells reported previously could be pinpointed to specific complexes or pathways (Tomas et al., 2017).

In myotubes only two of the substrate combinations produced viable results. The addition of pyruvate and malate, palmitoyl-l-carnitine, and glutamine and malate produced negative values for OCR in both CFS (n = 6) and control myotubes (n = 6) (Fig. 3). The addition of the injected compounds did appear to have an effect on the OCR, but not the anticipated effect, and only achieved the result of producing more negative OCR readings with the addition of each injection and not the expected increase after the first injection. These experiments were repeated a number of times with similar traces produced each time. This suggests that this technique for measuring mitochondrial complex activity in permeabilised cells may not be appropriate for use in myotubes. Successful recordings of the effect of succinate, and TMPD and ascorbate were made in control and CFS myotubes. No difference in state 3 respiration, state 4 respiration, or RCR were seen between the control and CFS cohorts. This indicates that there are no abnormalities in respiratory activity linked to glutaminolysis, or complex IV activity in CFS myotubes. This is in agreement with previous research which, using 3 carboxyl-14C–labelled substrates, found there to be no difference between CFS and control skeletal muscle cell complex I, complex II + III, complex III, or complex IV activity (Smits et al., 2011). However, given the inconsistency of between substrate readings in this study, with some of the substrates giving successful readings and some not, and the large error bars shown on the traces for each of the substrates (including the substrates for which we could successfully derive state 3 & 4 respiration), the use of this technique to accurately record the activity of specific aspects of mitochondrial respiration in permeabilised myotubes should be questioned. Other techniques such as phosphorescence oxygen sensitive probes and spectrophotometric enzyme assays should be used to analyse the same samples in future experiments in order to determine if these techniques can provide more accurate results than those achieved here with the XFe96 and to see the consistency between techniques. Very few studies have previously been published using extracellular flux analysis to detect mitochondrial activity in permeabilised myotubes. One study used extracellular flux analysis and high resolution respirometry to determine the differences between the techniques (Boyle et al., 2011). The study showed traces from the Seahorse XFe24 to have higher levels of variability for each data point which supports the data reported here showing that extracellular flux analysis may not be the most suitable technique for detecting changes in mitochondrial complex activity in permeabilised myotubes.

In PBMCs, successful traces were recorded for all five combinations of substrates. State 3 respiration, state 4 respiration, and RCR of control (n = 6, succinate controls n = 4) and CFS (n = 13) PBMCs were compared. No difference was seen between the two cohorts in any of the substrates investigated. This suggests that the activity of complexes I, II and IV, and respiration supported by fatty acid oxidation or glutaminolysis do not differ in CFS PBMCs compared with healthy controls. These results are in agreement with a study by Lawson et al. (2016) who used spectrophotometric techniques to show that permeabilised CFS PBMCs do not differ from healthy controls in terms of complex I, complex II–III, or complex IV activity. The consistency of results between the different research groups, using different techniques, strengthens the evidence suggesting that there are no abnormalities in individual mitochondrial complex activity in CFS PBMCs. The lack of differences in PBMCs may suggest that the abnormalities found in whole cells shown previously is not due to abnormalities in the mitochondrial respiratory chain complexes but rather at different points of the respiration pathway such as movement of glucose into cells, AMPK abnormalities, or altered functioning of other mitochondrial enzymes  (Tomas et al., 2017). However, caution must be used in interpreting these results on their own given the differences in OXPHOS observed in whole PBMCs (Tomas et al., 2017), as the results from whole cells reflect a more natural and physiologically relevant environment for the mitochondria. While relatively small sample sizes were used in this study, the consistency of the findings of this study with previous studies using different techniques to measure mitochondrial complex activity in myotubes and PBMCs in CFS patients adds validity to the results (Lawson et al., 2016; Smits et al., 2011).

Conclusions

This study investigated the activity of specific components of mitochondrial respiration by looking at individual complex activity and pathways in myotubes and PBMCs in a case-control study. A cell permeabilization protocol developed by Salabei et al. using the XFe96 extracellular flux analyser was used to conduct these experiments. This is the first study to use extracellular flux analysis to investigate individual mitochondrial complex activity in permeabilised cells in the context of CFS. Ultimately, normal mitochondrial function was recorded in CFS myotubes and PBMCs; however, relatively small sample sizes were used therefore the results should be interpreted with caution. The finding of normal mitochondrial functioning of CFS myotubes supports the results of unpublished data from whole cells (Rutherford, 2016). The results showing no difference in mitochondrial activity in permeabilised PBMCs were unexpected given that mitochondrial function in PBMCs has previously been shown to be significantly lower in CFS (Tomas et al., 2017). However, the lack of difference in complex activity in CFS PBMCs is in agreement with results reported by other groups who showed normal mitochondrial respiratory chain complex activity (Lawson et al., 2016; Vermeulen et al., 2010), and postulated that changes in mitochondrial ATP synthesis should be attributed to other causes such as the transport capacity of oxygen (Vermeulen et al., 2010). Given the results here, the future of bioenergetic studies in CFS should concentrate on mechanisms upstream of the mitochondrial respiratory chain.

Supplemental Information

Data S1 Raw data

The raw measures are provided in Supplementary File 1. The raw data shows state 3 respiration, state 4 respiration, and the respiratory control ratio (RCR) for PBMCs and myotubes from ME/CFS patients and healthy controls.

Click here for additional data file.

The authors would like to thank all the patients and control subjects who participated in this study.

Additional Information and Declarations

Competing Interests

Author Contributions

Ethics

Data Availability

Joanna L. Elson is an Academic Editor for PeerJ.

Cara Tomas conceived and designed the experiments, performed the experiments, analyzed the data, prepared figures and/or tables, authored or reviewed drafts of the paper, approved the final draft.

Audrey E. Brown conceived and designed the experiments, performed the experiments, authored or reviewed drafts of the paper, approved the final draft.

Julia L. Newton conceived and designed the experiments, contributed reagents/materials/analysis tools, authored or reviewed drafts of the paper, approved the final draft.

Joanna L. Elson conceived and designed the experiments, analyzed the data, contributed reagents/materials/analysis tools, authored or reviewed drafts of the paper, approved the final draft.

The following information was supplied relating to ethical approvals (i.e., approving body and any reference numbers):

Blood samples were obtained from ME/CFS patients after obtaining ethical approval from the National Research Ethics Committee North East—Newcastle & North Tyneside 2. Samples from healthy controls were collected through the Institute of Cellular Medicine (Newcastle University) blood study after obtaining ethical approval from the National Research Ethics Committee North East—County Durham & Tees Valley. REC references: 12/NE/0146; 12/NE/0121.

The following information was supplied regarding data availability:

The raw measures are available in Supplemental File.

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
