# Peer review of "Mitochondrial complex activity in permeabilised cells of chronic fatigue syndrome patients using two cell types"

_PeerJ, doi:10.7717/peerj.6500_

## Round 0.1 · original submission · Minor Revisions

Dear Dr. Tomas and colleagues:

Thanks for submitting your manuscript to PeerJ. I have now received two independent reviews of your work, and as you will see, both are very favorable. Well done! Nonetheless, both reviewers raised some relatively minor concerns about the research, and areas where the manuscript can be improved. I agree with the reviewers, and thus feel that their concerns should be adequately addressed before moving forward.

Therefore, I am recommending that you revise your manuscript accordingly, taking into account all of the issues raised by the reviewers. I do believe that your manuscript will be ready for publication once these issues are addressed.

Good luck with your revision,

-joe

Reviewer 1 ·

Basic reporting

The paper is generally sound. There are however several typos in the document which need to be corrected.
112 missing word: Diagnostic
147 Possible typo EGTA
154 Typo injectizon
249 Typo extra work ‘for’
104 Workd- work

References are OK. One is a PhD thesis reference 30

Experimental design

120 General comment: No information regarding suppliers of materials
126 How was differentiation of myoblast to myotubes established?
132 Were the wells pre-coated with poly-d-lysine? This may impact cell survival
205 In table 1, the authors state that glutamine was injected however the figure 4 states glutamate. The authors must confirm which compound was used.

I wonder whether the title of the paper is a true reflection of the data?

Validity of the findings

The permeabilised myoblast expt failed to work likely due to insufficient cell/mitochondria.

Not too much should be read into the myoblast data.

239 Successful recordings of the effect of glutamate and malate,
and TMPD and ascorbate were made in control and CFS myotubes Glutamate does not appear to work in myoblasts (Fig3A). is this statement correct?

Glutamate and Glutamine are mixed up. The authors need to make sure the correct word is used.

205 In table 1, the authors state that glutamine was injected however the figure 4 states glutamate. The authors must confirm which compound was used.

Did the myoblasts and PBMC used the same glutamine/glutamate substrate

Additional comments

Although on the small side this is a useful study carried out using similar approaches to previous work by the group on PBMCs. The limitations of myoblasts in this type of approach are mentioned and the authors should not over interpret the data based on the myoblasts.

The PBMC data is solid showing no difference between patients and controls with respect to mitochondrial complex activity

One thing to consider for future expts is the mitochondrial burden of pen and strep in conjunction with antifungal amphotericin B. Pen and strep reduces OCR in human fibroblasts. It would be worth investigating how these anitbiotics and fungicides affect the results particularly on PBMCs.

Reviewer 2 ·

Basic reporting

The paper is well written and most of the references are appropriately cited. The authors have covered the literature and background of the study logically and appropriately. The language of the manuscript explaining results and background reads elegant. The scientific findings are clearly explained in professional English throughout the manuscript. The article conforms to an acceptable format of the journal. Figures legends are well described, and figures are of acceptable quality. However, in line 63-68 quote “Previous studies have shown the energy production, including mitochondrial activity, of whole PBMCs from CFS patients to be significantly impaired compared to a healthy control cohort (4, 15). One study showed CFS PBMCs to have significantly impaired mitochondrial functioning both under basal conditions and when maximally stimulated to respire, under a number of experimental conditions (4).” I would suggest authors to slightly elaborate on the technique used in previous study to draw the conclusion of defective mitochondrial function. Then it will bring the contrast of technological limitation of previous findings and importance of extracellular flux assay as used in the current study.

Experimental design

The present study has clearly and unambiguously laid down the research question and the experiments are well designed to support the claims. The study investigated a plausible mitochondrial dysfunction as a cause of CFS. Authors explored this hypothesis using extracellular flux assay using seahorse. It serves as a great tool to fill in the gap on functional role of mitochondrial ETC complex in the pathogenesis of CFS. The method section has been well described for any independent investigator to repeat the study.

Validity of the findings

Authors have taken good care to include proper control in their data representation. The conclusions in the present study is appropriately stated and authors themselves are cautious in making far stretched conclusion of the study, which is impressive. Small sample sized of the study is also well acknowledged by the authors while drawing conclusion from the experiments.

Additional comments

Though the study shows a negative data, but the authors in the study have used all proper controls to draw their conclusion. I will recommend this work to be accepted for publication with no revision. One thought I ahd was, it would have been nice if authors had used some positive control e.g. say cells with defect in complex 1 activity in their seahorse study to at least show that the assay is working but their experimental setup do not show any positive effects between control and CFS derived myotubes/PBMC.

---

## Round 0.2 · accepted · Accept

Dear Dr. Tomas and colleagues:

Thanks for revising your manuscript based on the concerns raised by the two reviewers. I now believe that your manuscript is suitable for publication. Congratulations! I look forward to seeing this work in print, and I anticipate it being an important resource for communities studying chronic fatigue syndrome and mitochondrial function. Thanks again for choosing PeerJ to publish such important work.

Best,

-joe

#